# Omitting Sentinel Lymph Node Biopsy after Neoadjuvant Systemic Therapy for Clinically Node Negative HER2 Positive and Triple Negative Breast Cancer: A Pooled Analysis

**DOI:** 10.3390/cancers15133325

**Published:** 2023-06-24

**Authors:** Munaser Alamoodi, Umar Wazir, Kinan Mokbel, Neill Patani, Jajini Varghese, Kefah Mokbel

**Affiliations:** 1Faculty of Medicine, King Abdulaziz University, Jeddah 21589, Saudi Arabia; malamoodi@kau.edu.sa; 2The London Breast Institute, Princess Grace Hospital, London W1U 5NY, UK; umar.wazir@rcsed.ac.uk (U.W.); k.mokbel@ex.ac.uk (K.M.); neill.patani@hcahealthcare.co.uk (N.P.); jajini.varghese1@nhs.net (J.V.); 3Department of Surgery, Khyber Teaching Hospital, Peshawar 25120, Pakistan; 4College of Medicine and Health, University of Exeter Medical School, Exeter EX1 2LU, UK; 5Department of General Surgery, University College London Hospital, Euston Road, London NW1 2BU, UK; 6Department of General Surgery, Royal Free Hospital, London NW3 2QG, UK

**Keywords:** breast cancer, neoadjuvant systemic chemotherapy, sentinel lymph node biopsy, complete pathological response, complete radiological response, clinically node negative, pathological node negative, pathological node positive

## Abstract

**Simple Summary:**

Following neoadjuvant systemic therapy (NAST), patients who were clinically node-negative at diagnosis still routinely undergo sentinel lymph node biopsy (SLNB) to detect nodal disease. Surgical staging of the axilla is currently the standard of care, including for those who achieve complete imaging response in the breast and/or axilla. It has been well established that certain breast cancer subtypes respond better to NAST and are more likely to achieve pathological node-negative status (ypN0). These complete responses are underpinned by advances in systemic therapy and subtype-specific targeted treatment. Our pooled analysis shows that patients with no clinical evidence of axillary node involvement at diagnosis, who respond well to upfront systemic therapy, have around 2% chance of disease in sentinel lymph nodes. This suggests that where the risk of nodal disease is sufficiently low, there is a possibility of safely omitting axillary surgery in selected patients.

**Abstract:**

Recent advances in systemic treatment for breast cancer have been underpinned by recognising and exploiting subtype-specific vulnerabilities to achieve higher rates of pathologic complete response (pCR) after neo-adjuvant systemic therapy (NAST). This down-staging of disease has permitted safe surgical de-escalation in patients who respond well. Triple-negative (TNBC) or HER2-positive breast cancer is most likely to achieve complete radiological response (rCR) and pCR after NAST. Hence, for selected patients, particularly those who are clinically node-negative (cN0) at diagnosis, the probability of disease in the sentinel node after NAST could be low enough to justify omitting axillary surgery. The aim of this pooled analysis was to determine the rate of sentinel node positivity (ypN+) in patients with TNBC or HER2-positive breast cancer who were initially cN0, achieving rCR and/or pCR in the breast after NAST. MedLine was searched using appropriate search terms. Five studies (N = 3834) were included in the pooled analysis, yielding a pooled ypN+ rate of 2.16% (95% CI: 1.70–2.63). This is significantly lower than the acceptable false negative rate of sentinel lymph node biopsy (SLNB) and supports consideration of omission of SLNB in this subset of patients.

## 1. Introduction

Surgical decision-making is a responsive process, taking into account patient factors and tumour biology and carefully balancing the potential risks and benefits of each intervention. In the context of breast cancer, the evolution of surgical practice has resulted in gradual de-escalation. Historically, a modified radical mastectomy with axillary lymph node dissection (ALND) was considered the gold standard for early breast cancer treatment. However, as the mechanistic model of breast cancer was superseded by a systemic model based on tumour biology, radical surgery has also been replaced by more tailored and targeted procedures such as wide local excision (WLE) with radiotherapy and sentinel lymph node biopsy (SLNB). These changes in surgical practice were underpinned by landmark studies and meta-analyses of randomised controlled trials comparing treatments and equivalence of outcomes [1].

The routine de-escalation of axillary surgery in early breast cancer from ALND to the current practice of SLNB has significantly reduced morbidity and improved patient quality of life without compromising oncological outcomes. With the help of national screening programs, cancers are being detected earlier, and patients are increasingly diagnosed with clinically undetectable lymph node disease (cN0). There are subgroups of patients now recognized to be at sufficiently low risk of nodal disease that routine sentinel node biopsy may also be unnecessary. The American Society of Clinical Oncology (ASCO) and Cancer CARE Ontario now recommend the omission of SNLB in patients over 70 years with T1 cN0 invasive cancer that is HR-positive and HER2-negative [2]. This guideline for patients over 70 years with low-risk breast cancer is supported indirectly by the CALGB 9343 trial. Here, nearly two-thirds of trial patients had no surgical staging of the axilla, and there were no axillary recurrences among those receiving adjuvant radiotherapy and endocrine therapy [3]. There has been much data recently published comparing the morbidity of patients undergoing SLNB and those being followed up by ultrasound. The SOUND study recently presented at the St. Gallen Breast Cancer Conference is one such study [4]. Furthermore, the INSEMA study has very clearly shown better breast and arm-related symptoms in patients in whom SLNB was omitted in comparison to those in whom it was done [5].

The phenomenon of surgical de-escalation continues in several domains, including relatively recent interest in the safety of omitting SLNB in selected patients after neoadjuvant systemic therapy (NAST). The standard of care for patients receiving NAST currently recommends surgical staging of the axilla after treatment, including those who were clinically node-negative at diagnosis. However, with advances in systemic therapy, particularly subtype-specific treatments, this practice of routine sentinel biopsy is currently the subject of much debate. The two patient populations most likely to respond completely to NAST are those with HER2-positive disease receiving dual anti-HER2 targeting and triple-negative breast cancer (TNBC) receiving immunotherapy, which has been shown to increase the rate of pCR and ypN0 [1,6]. Within these favourable subtypes, patients who were clinically node-negative (cN0) and achieved pathologic complete response (pCR) in the breast are most likely to have no axillary nodal disease after treatment (ypN0). Hence, strict post-NAST assessment of response using magnetic resonance imaging (MRI) and axillary ultrasound could potentially identify cN0 HER2-positive and TNBC patients, where the morbidity of sentinel node biopsy could be avoided without detriment to local disease control or long-term oncological outcome.

## 2. Materials and Methods

The Medline database was searched using PubMed with the following key terms: pCR and NAST, HER2+ and NAST, pCR and cN0, TNBC and NAST. Articles returned by the search were assessed according to the following inclusion criteria: HER2+ and TNBC with cN0 pre-treatment and pCR with ypN0. The databases were accessed in May 2023, and no limit was placed on the time of publication of the studies.

Pathologic complete response in the breast was defined as no invasive or in situ cancer (ypT0), based on the German Breast Group (GBG) definition of pCR in the neoadjuvant therapy setting [7].

Articles were discarded from the pooled analysis based on the following exclusion criteria: (1) Abstracts, case reports, and lectures, (2) Results for cN1 post-NAST, (3) Incomplete data that were unable to be extracted from other relevant studies.

Articles that met the aforementioned criteria were assessed by two reviewers, and data were extracted and compiled for analysis. Statistical analysis involved using standard deviation with normal approximation to the binomial calculation.

## 3. Results

The literature search identified a total of 31 studies, out of which five studies met the predefined inclusion criteria (Figure 1). These selected studies encompassed a patient population of 3834 individuals who fulfilled the specified selection criteria [8,9,10,11,12]. Table 1 presents the tabulated information, including the author names, the number of patients (N) with cN0 TNBC or Her2 positive breast cancer who achieved pCR after NAST and frequency of nodal disease post-treatment (ypN+).

Analysis of final pathology following NAST revealed an overall frequency of nodal disease (ypN+) of 2.16% (95% CI:1.70–2.63). This finding highlights the presence of residual disease in a small proportion of patients after NAST.

For patients with TNBC who exhibit residual disease following NAST, capecitabine has demonstrated its effectiveness in improving clinical outcomes. In a landmark study, a reported 8.5% overall survival benefit was observed at the five-year mark [13]. Consequently, failing to identify residual disease in 2.16% of patients could potentially impact their outcomes. This impact is estimated to affect 1.8 out of every thousand patients (0.0216 × 0.085) who would miss the opportunity to receive adjuvant capecitabine due to the omission of SLNB.

Similarly, adjuvant Trastuzumab Emtansine (TDM-1) has shown efficacy in enhancing clinical outcomes for patients with HER2-positive breast cancer who retain residual disease after NAST [14]. The estimated benefit in terms of distant disease-free survival at three years was reported to be 5.4%. Consequently, failure to detect residual nodal disease in 2.16% of patients could potentially influence their outcomes, affecting an estimated 1.17 per thousand patients (0.0216 × 0.054) who would not receive adjuvant trastuzumab emtansine (TDM-1).

## 4. Discussion

This systematic review of the literature and pooled analysis demonstrated the extremely low nodal positivity rate (2.16%) in HER2-positive and TNBC after NAST in patients with initially cN0 disease who achieved pCR in the breast. This rate is consistent with the published literature of 2–3% (95% CI: 1.70–2.63) [9,15]. These data also demonstrate the strong correlation of pCR with ypN0 post-NAST in these two molecular subtypes. The current study adds to the growing debate regarding the necessity of performing SLNB in such patients. Notably, the incidence of sentinel node disease in such patients is significantly lower than the acceptable false-negative rate (FNR) of the SLNB procedure itself, which is not without risk of morbidity. Furthermore, the potential adverse impact of omitting SLNB, specifically the missed opportunity for subtype-specific adjuvant targeted treatment, has been estimated to be less than 1 in 1000 patients. The overall FNR of the SLNB was reported to be 8.61% (CI: 8.05–9.2%) according to a meta-analysis of published studies [16], and randomised controlled trials showed that this recognised FNR had no adverse impact on overall survival, disease-free survival and regional control [17].

Avoiding unnecessary SLNB in breast cancer patients can achieve savings of approximately USD $1500 per case [18]. Additionally, SLNB is associated with recognized complications, including lymphoedema, pain and motor and sensory disorders. The prevalence of lymphoedema following SLNB after 24 months is reportedly 5.9%. Furthermore, 17% of patients undergoing SLNB experienced a reduced range of movement of the arm following surgery [19]. At six months of follow-up, 11–16% reported persistent pain, while 2–22% and 0–9%, respectively, reported sensory and motor disorders [20]. Sentinel node mapping carries an anaphylaxis risk of 0.083% related to the use of blue dye [21]. Therefore, omitting SLNB in selected patients has substantial benefits related to quality of life and cost-effectiveness.

Patients with TNBC or HER2-positive disease are recognised to have the highest rates of pCR in the breast [1]. Tadros et al. have shown that pCR after NAST correlates with ypN0 [8]. Rates of breast pCR and ypN0 differ across tumor subtypes and are highest in HER2+ or TNBC [6]. This observation is also confirmed by van der Noordaa et al. [15]. Other predictors of achieving pCR include high tumor grade and high proliferation index. On the contrary, hormone-sensitive lobular breast cancer is associated with the lowest rate of pCR. However, accurate prediction of pCR remains challenging. The standard conventional imaging method for predicting pCR utilizes the combination of mammography with ultrasound [22]. However, breast magnetic breast imaging (MRI) seems to estimate the response of the primary tumor to NAST more accurately than conventional breast imaging [23], particularly in patients with HER2+ or TNBC (Figure 2 and Figure 3) [24]. Breast MRI has received the highest accuracy rating (rated 9) by the American College of Radiologists [25]. Furthermore, when considering patients with triple-negative or HER2-positive breast cancer who achieve a complete or partial response on breast imaging, the prediction of pCR can be improved by sampling the index breast area. Accuracy of 98% with an FNR of 5% has been achieved with vacuum-assisted core biopsy (9-gauge), obtaining 12 cores, in addition to fine needle aspiration cytology after NAST [26].

There are several prospective trials ongoing that specifically address the omission of axillary SLNB in cN0 patients with early breast cancer. Hersh and King published an overview of three European randomised trials investigating the omission of axillary SLNB in cN0 patients with early breast cancer (SOUND; INSEMA; and BOOG 13-08) and an Asian (NAUTILUS) adjuvant surgical trial [27]. These trials represent important milestones supporting the move to de-escalate SLNB in selected patients, and participation in these trials is strongly encouraged. However, one of the aims of omitting SLNB is to improve quality of life (QoL). Although; none of the current trials have QoL as a primary endpoint, the expectation is that a decrease in surgical procedures will have a positive impact [28]. The European trials (EUBREAST-01; ASICS) include only patients with the highest likelihood of achieving pCR after NAST (HER2+ and TNBC), which should support safely omitting SLNB in carefully selected patients; alongside the randomised trials detailed above [29,30].

The optimal timing of SLNB in the context of NAST also remains a topic of debate, with different guidelines making different recommendations. The 2014 ASCO guidelines suggested that SLNB could be performed either before or after NAST in cN0 women with operable breast cancer [31]. However, the updated German S3-guidelines recommend performing SLNB after NAST for patients with clinically and sonographically node-negative pre-treatment status [32]. As more evidence emerges, it is likely that these guidelines will be revised to reflect the evolving evidence that SLNB can be safely omitted in certain patients who respond well to NAST. Specifically, patients with cN0 HER2+ and TNBC who achieve pCR after NAST may be able to forego SLNB altogether. Furthermore, advances in systemic therapy for breast cancer have led to higher rates of pCR among patients receiving NAST, providing an opportunity to reduce or even eliminate breast surgery in these patients [1,33].

Barron reported that in patients with cN0 HER2+ disease or TNBC with breast pCR, the nodal positivity (ypN+) rate was 1.6% for both subtypes, which is in keeping with our findings. Rates of ypN+ were higher in patients with cN0 and residual disease in the breast (16.9% in HER2+ and 12.6% in TNBC). Hence patients with residual breast disease post-NAST have an appreciably higher rate of ypN+ despite being cN0 pre-treatment [9]. Samiei reported similar findings [34]. Therefore, it is important to continue recommending SLNB as the best practice for such patients to optimize adjuvant treatment and clinical outcomes.

Although our study is the first to conduct a combined analysis of the international literature with a sample size exceeding 3000 patients, it has several limitations. Firstly, the studies included in our analysis were retrospective and heterogeneous, lacking standardized protocols for the definition of cN0, radiological and pathological complete response. It is also noteworthy that the SLNB technique was not standardised. The data regarding pCR were derived from surgical specimens obtained during the same surgical procedure when the SLNB was performed. Finally, most of the data were extracted from a single study, which may introduce bias (Barron et al.) [9]. Unfortunately, due to the heterogeneity of the studies, we were unable to perform subgroup analyses.

The data presented here should assist patients in making informed decisions about the surgical staging of the axilla. Carefully selected patients who achieve a complete radiological response on breast MRI after NAST should be offered the option of omitting SLNB after a multi-disciplinary discussion. However, if the final pathology reveals residual breast disease, then SLNB should be undertaken as a second procedure to avoid under-staging of residual axillary disease. Patients undergoing total mastectomy may benefit from receiving an interstitial injection of super-paramagnetic iron oxide (SPIO) during the initial breast surgery to facilitate subsequent SLNB if deemed necessary [35]. We strongly encourage participation in the ongoing EUBREAST-01 multi-center clinical trial, which is currently addressing this topic [30]. The results of which are likely to have a significant positive impact on patient’s quality of life and healthcare cost-effectiveness through further safe de-escalation of axillary surgery.

## 5. Conclusions

Patients with clinically node-negative (cN0) HER2-positive or TNBC, who achieve complete response after neoadjuvant systemic therapy (NAST), have a remarkably low incidence of sentinel node positivity after treatment, which is less than the false-negative rate of sentinel node biopsy itself. Hence, for carefully selected patients, the option of safely omitting sentinel biopsy should be discussed by multi-disciplinary teams and represents an important aspect of informed consent currently lacking. Patients should be informed that omitting SLNB can avoid significant morbidity and is highly unlikely to have any detrimental impact on long-term oncological outcomes.

## Figures and Tables

**Figure 1 cancers-15-03325-f001:**
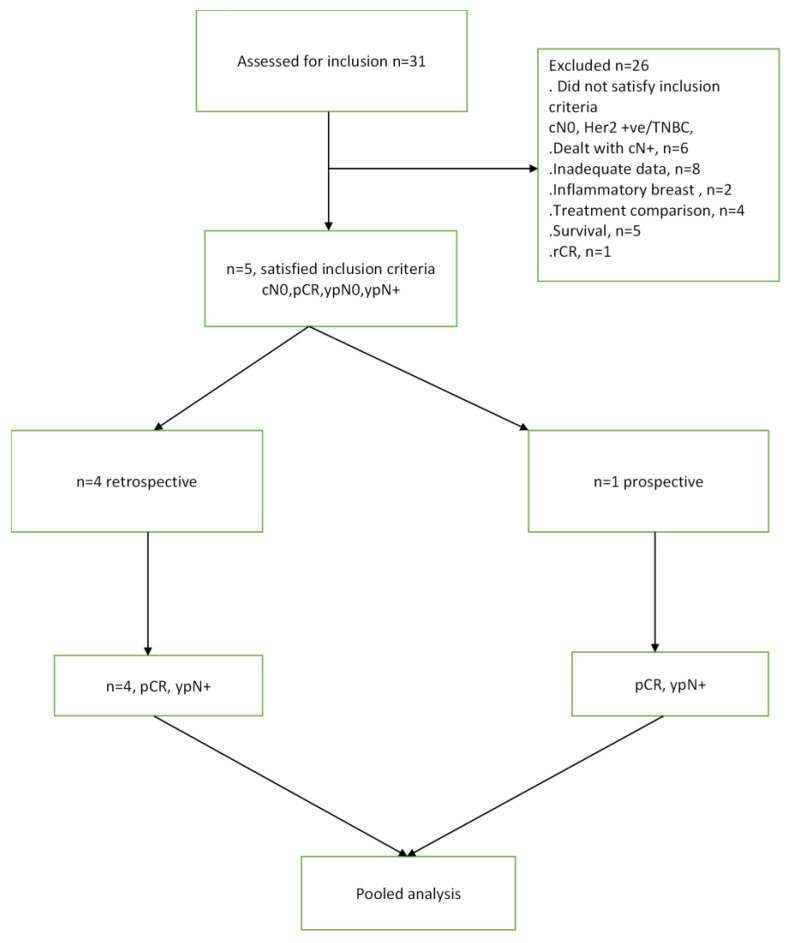
CONSORT diagram for this study.

**Figure 2 cancers-15-03325-f002:**
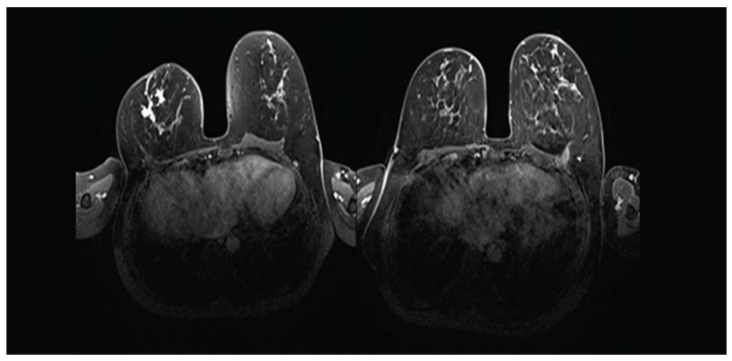
MRI demonstrating complete radiological response of a TNBC in the right breast after NAST that included Carboplatin and Pembrolizumab in a 50-year-old woman (left: before NAST; right: after NAST). The patient achieved pCR.

**Figure 3 cancers-15-03325-f003:**
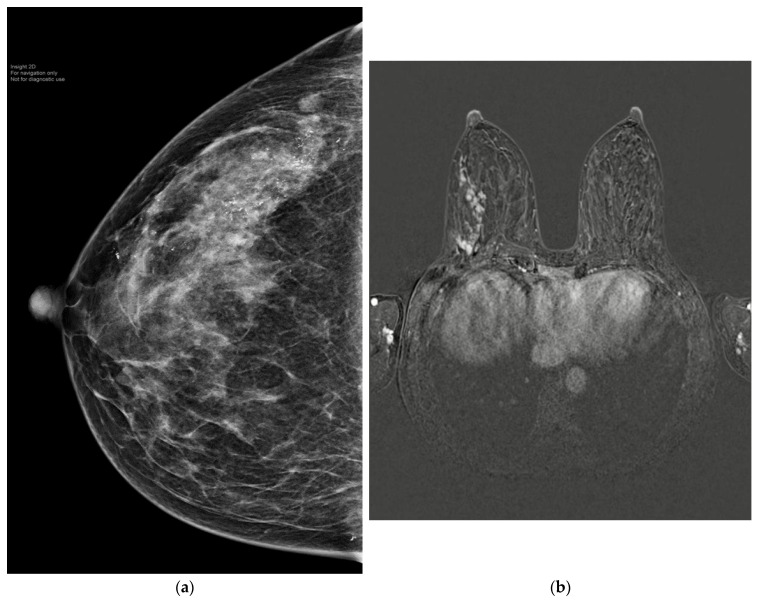
(**a**) A 50-year-old patient with extensive HER2-positive breast cancer in right upper outer quadrant. (**b**) MRI showed extensive mass and non-mass-like enhancement spanning 10 cm. The patient received neoadjuvant weekly Paclitaxel and Herceptin with Pertuzumab for three months. (**c**) post-NAST MRI showed a complete radiological response. The patient underwent nipple-sparing mastectomy and reconstruction with SLNB. The final histology confirmed pCR.

**Table 1 cancers-15-03325-t001:** Studies included in the systematic review and pooled analysis. pCR: complete pathological response to neoadjuvant therapy; ypN+: pathologically node-positive disease after neoadjuvant systemic therapy (NAST).

Study	Citation	Study Design	Number of Patients (N)	ypN+ Cases (%)	Comments
Tadros A.B et al.	[8]	Prospective cohort	116	0 (0)	pCR
Barron et al.	[9]	Retrospective	3240	75 (0.03)	pCR
Zhi-Qiang et al.	[10]	Retrospective	17	0 (0)	pCR
Samiei et al.	[11]	Retrospective registry	353	4 (1.1)	pCR
Weiss et al.	[12]	Retrospective	108	4 (3.7)	pCR
Total			3834	83 (2.16)	Overall ypN+ = 2.16%

## Data Availability

The data presented in this study are available in this article.

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
