# Peer review of "Omitting Sentinel Lymph Node Biopsy after Neoadjuvant Systemic Therapy for Clinically Node Negative HER2 Positive and Triple Negative Breast Cancer: A Pooled Analysis"

_cancers, 2023, doi:10.3390/cancers15133325_

Round 1
Reviewer 1 Report
Overall a very well written and useful analysis. Comments are:
Question is whether this is sufficient for clinical practice, whether it is sufficient for an RCT or neither.
The measure used to assess ‘respond well to upfront systemic therapy,’ is important.
Abstract:
Abstract is more more intro than anything, could shorten the intro component and give a little more detail of the results.
Rate of positive LNs in patients at 2.1% is certainly low and supports the hypothesis that SLNBx may be omitted in this group.
Intro
Agree if clinical LN negative (presume by US/MRI), TNBC or HER2 and good response, chance of LN involvement unlikely.
Agree also, rate of axillary recurrence is surprisingly low in cN0 low risk breast cancer. Additionally in trials omitting clearance in positive SLNs axillary relapse axillary recurrence rates are also very low.
Agreeably surprised number of studies and size – N=3941
Methods
Pooled analysis appropriate to deal with available type of data.
Would also be good to see a CONSORT diagram detailing reason as for exclusion of 25 of 31 studies identified
Results
What proportion of these patients were HER2 positive and TNBC and did the positive SLNBx rate vary between the two. This is covered in the discussion for the Barron paper but could go in Table 1 if available for most studies.
HER2 positive cancers are divided about 50:50 into HER2 pos/ER neg and HER 2pos/ER pos – pCRs are lower in ER pos disease. Did data allow a comparison of pos SNLBx after pCR between these two groups.
pCR is obviously rare in ER pos HER2 neg but would still be interesting to know if SLNBx can be omitted there also though appreciate would be excessive work to extract if not already available. Could mention these patients in discussion otherwise.
I would question the value of including radiological rCR in this study as it appears to be one study of 107 patients, especially as many patients receiving NAST do not receive an MRI post-therapy in routine clinical practice so it would be harder to apply unlike pCR. I would tend to include just pCRs and state that omitting pCR appears valid in pCR cases but mention the Van der Noorda study in discussion as the authors already have and that this could extend to rCRs.
Note Table 1 has a typo – Barron et et al – third line 75 of 3240 cases is 2.3% not 23%.
Calculations regarding omitting capecitabine (as per CreateX) and T-DM1 (as per the Catherine trial) are useful. The actual excess risk of mortality is obviously substantially lower again than these therapy omission figures.
Discussion
Generally very good. Covers the literature in general, the validity of rCR as a predictor and ongoing trials.
Could mention what situation is for ER pos/HER2 neg pts with pCR, accepting pCR rare.
Limitations well stated.
Author Response
Thank you for your comments. We have applied your recommendations as well as possible, and believe that they have improved the manuscript greatly.
- "...could shorten the intro component and give a little more detail of the results.": We have applied this.
- "Would also be good to see a CONSORT diagram detailing reason as for exclusion of 25 of 31 studies identified": Done
- "I would question the value of including radiological rCR in this study as it appears to be one study of 107 patients": We have removed this study.
- "Note Table 1 has a typo – Barron et et al – third line 75 of 3240 cases is 2.3% not 23%": Corrected.
- "HER2 positive cancers are divided about 50:50 into HER2 pos/ER neg and HER 2pos/ER pos – pCRs are lower in ER pos disease. Did data allow a comparison of pos SNLBx after pCR between these two groups.": It is an interesting point, but we were unable to extrapolate an answer to it on the basis of the information we seen.
Reviewer 2 Report
The work presented here deals with a highly topical issue that is currently being investigated in clinical trials. The de-escalation of axillary surgery for breast cancer is one of the most frequently discussed topics regarding the therapy of breast cancer. However, some minor changes or revisions are needed so that the topic can be discussed with the most current data:
1. The introduction should mention the data from the SOUND study recently presented at the St. Gallen Breast Cancer Conference. Furthermore, a statement on morbidity rates after SLNB and mention of the recently published data comparing arm morbidity from the INSEMA study would be recommended (Reimer et al. 2023) to emphasize the advantage of omitting SLNB.
2. Under Material and Methods, the time period of the literature search should still be specified (studies from which publication period were included?).
3. Under results, the design of the included studies should be described in more detail: pro- or retrospective, registry studies or interventional studies, proportion of pat. with breast-conserving therapy and mastectomy, imaging performed to determine cCR. Was the GBG definition of pCR used in all studies?
4. In Table 1, please insert the reference numbers.
5. The conclusion that SLNB can be dispensed with depending on MRI does not seem correct to me. Not everywhere is breast MRI standard for monitoring under NAST and certainly not always performed in the included studies.
Author Response
Thank you for your contributions. We have addressed your points as follow:
- We have added this to the Introduction.
- The following was inserted into Material & Methods: The databases were accessed in May 2023, and no limited was placed on the time of publication of the studies.
- We have inserted the information regarding study design. Whilst we cannot determine with certainty, we believe that GBG definition was the most widely used definition in these studies.
- Reference numbers inserted.
- In the conclusion, we have removed mention of MRI.